# *Helicobacter pylori* Infection in Patients with Gastric Cancer: A 2024 Update

**DOI:** 10.3390/cancers16111958

**Published:** 2024-05-22

**Authors:** Claudia Burz, Vlad Pop, Ciprian Silaghi, Iulia Lupan, Gabriel Samasca

**Affiliations:** 1Institute of Oncology “Prof. Ion Chiricuta”, 400015 Cluj-Napoca, Romania; cristina.burz@umfcluj.ro (C.B.); pop.vlad.vasile@elearn.umfcluj.ro (V.P.); 2Department of Immunology, Iuliu Hatieganu University of Medicine and Pharmacy, 400162 Cluj-Napoca, Romania; 3Department of Biochemistry, Iuliu Hatieganu University of Medicine and Pharmacy, 400349 Cluj-Napoca, Romania; silaghi.ciprian@umfcluj.ro; 4Institute for Interdisciplinary Research in Bio-Nanosciences, 400271 Cluj-Napoca, Romania; iulia.lupan@ubbcluj.ro; 5Department of Molecular Biology, Babes-Bolyai University, 400084 Cluj-Napoca, Romania

**Keywords:** gastric cancer, *Helicobacter pylori*, update, 2024

## Abstract

**Simple Summary:**

The primary risk regarding *Helicobacter pylori* infection is that it is associated with gastric cancer. The lack of eradication of *Helicobacter pylori* has led to an increase in the prevalence of gastric cancer. We examined the literature from the year 2024 related to the *Helicobacter pylori*/gastric cancer association. Immunological studies increased, but they did not convince in the study of this infection. Most of the examined studies address the clinical treatment of this bacterium. However, to the same extent, other studies have investigated genetically individualized treatment. If, until now, clinical treatments have not yet been able to eradicate this bacterium, we recommend genetically individualized treatment.

**Abstract:**

Numerous studies have been performed on *Helicobacter pylori* infection because of the high death rate linked to this illness and gastric cancer. An update on the key developments in recent years in the investigation of *Helicobacter pylori* and gastric cancer is the goal of this review. Using the search term “*Helicobacter pylori*, gastric cancer”, the PubMed database was searched. Only papers published in 2024 fulfilled the inclusion criteria. Because case report papers were not part of our investigation, they satisfied the exclusion criteria. Most of the research on the variable genes of *Helicobacter pylori* is guided by genetics to determine potential treatments. Studies on clinical treatments for the eradication of *H. pylori* with promising therapeutic options are needed. We found the fewest studies related to the immunopathology of *H. pylori* infection, which is still unknown. In conclusion, priority should be given to this kind of research.

## 1. Introduction

The identification of *Helicobacter pylori* (*H. pylori*) was originally described in the medical literature forty years ago. Subsequently, substantial advancements in both basic and clinical science have been implemented in routine clinical practice [1]. Nearly 50% of the global population is infected with this pathogen. The greatest concern about this infection is its direct association with gastric cancer (GC), which is the third most common cause of cancer-related death globally [2]. The majority of gastric malignancies are thought to be caused by persistent *H. pylori* infection. The exact processes that lead to the development of GC, such as the emergence of precancerous lesions and gastric polyps, are yet unknown [3]. In the last 20 years, there has been an increase in interest in *H. pylori* and GC research. Japan and China set standards and made the greatest contributions to this analysis [4]. *H. pylori*, obesity, and smoking are the primary risk factors that contribute to oncogenesis. It is also critical to make advancements in early detection, socioeconomic development (including improved drinking water, hygiene, and public sanitation), and nutritional practices [5]. Approximately 95% of stomach tumors in people older than 61 years are adenocarcinomas, the most common histological type of cancer [6]. *H. pylori* infection is more common in crowded houses and poor socioeconomic class households, especially in the noncardia subtype of stomach cancer. The development of GC is significantly influenced by diet, since certain eating patterns can irritate the stomach mucosa [7]. Poor lifestyle choices exacerbate the procarcinogenic epigenetic modifications caused by *H. pylori* exposure. Therefore, incorporating lifestyle, environmental, and epigenetic data into the stratification of primary gastric neoplasia risk can improve its resolution [8].

Despite colonizing the stomach epithelium, *H. pylori* infection has also been linked to several extradigestive tract conditions, such as colorectal cancer (CRC). Although an approximately twofold increased risk of CRC has been reported in epidemiological studies, direct causal and functional linkages between chronic infection and CRC have only recently been discovered [9]. Among patients with GC, CRC is thought to be the most common synchronous malignancy. It is essential to develop screening protocols that focus on high-risk categories to prevent CRC and lessen the disease burden [10]. From the beginning of *H. pylori* infection until the progression of CG, gastric epithelial barrier therapy is ineffective. After the “point of no return”, many pathways may be responsible for the ensuing carcinogenesis processes [11]. To determine the presence of *H. pylori* infection, more sensitive and thorough diagnostic techniques must be used, and any other variables that can affect the course of treatment must be carefully considered. Consequently, a more complex understanding of the relationship between GC and *H. pylori* infection is required [12,13]. This review aims to provide an update on the major developments in the study of *H. pylori* and GC in recent years. The term “*Helicobacter pylori*, gastric cancer” was used to search the PubMed database. Only articles from 2024 met the inclusion requirements. The case report articles met the exclusion criteria because they were not included in our investigation.

## 2. Immunopathology and Virulence Factors

Host immune response to *H. pylori* infection was studied in the medical literature (Table 1). An immunosuppressive environment and localized inflammation are caused by *H. pylori* infection. Elevated recruitment and induction of regulatory T cells (Tregs) in the GC tumor microenvironment (TME) are linked to high Treg infiltration; the precise role of infiltrated Tregs in GC is also influenced by immunosuppressive chemicals and phenotypes. The induction, recruitment, and activity of tumor-associated macrophages (TAMs) in the TME of GC are also regulated by several variables. TAMs facilitate the growth and spread of malignancies [14]. Throughout *H. pylori* infection, both in vitro and in vivo, the levels of Toll-like receptor 6 (TLR6) and inflammatory cytokines increased and finally decreased. The human gastric epithelial cell (GES)-1 production of interleukin (IL)-1β and IL-8 was enhanced by the restoration of TLR6, which attracted neutrophils and decreased the amount of *H. pylori* colonizing the stomach mucosa of gerbils. The mechanism by which continuous *H. pylori* infection decreases TLR6 sensitivity to bacterial components and controls inflammatory cytokine production in GES-1 cells via TLR6/Jun N-terminal kinase (JNK) signaling has been revealed. Both in vitro and in vivo, a TLR6 agonist reduced inflammation [15].

During *H. pylori* infection, forkhead box P3+ (FOXP3+) T cells detect ambient inflammatory signals such as IL-6, which is responsible for the immune response and exacerbates gastritis. By stimulating nuclear factor kappa B (NF-κB) signaling through the virulence component cytotoxin-associated antigen A (CagA), *H. pylori* elevates the expression of IL-6 in dendritic cells and macrophages, which in turn promotes the expression of IL-17A in FOXP3+ T cells [16]. Due to the difference in overall survival for *H. pylori* patients, many hypotheses have been proposed for trying to achieve a response. One possible explanation may be the population of tumor-infiltrating lymphocytes present in the tumor microenvironment. This matter was studied in xenograft models immunized with DNA vaccines. An increase in the ratio of cluster of differentiation (CD) 8+/CD4+ T cells reduced invasion by FOXP3+ Tregs. Moreover, the overexpression of caspase (cysteine-aspartic protease)-9/caspase-3, along with the downregulation of Survivin (a baculoviral inhibitor of apoptosis repeat-containing 5 belonging to the inhibitor of apoptosis (IAP) family), increased apoptosis [17].

*H. pylori* is known to exhibit virulence through three principal mechanisms: urease, vacuolating cytotoxin A, and the type IV secretion system (T4SS), which is synthesized by the cag pathogenicity island. Because the cag T4SS is specific to these gram-negative bacteria, potential endogenous proteins may inhibit *H. pylori* virulence, thus blocking the oncogenesis of GC. 1G2 is well known for its ability to act as a noncompetitive inhibitor of the adenosine triphosphate (ATP)ase activity of many other proteins. Consequently, in different studies, the efficacy of other inhibitors was evaluated by measuring the amount of IL-8 produced by GC cells infected with *H. pylori*. It has been determined that five molecules (1G2_1313, 1G2_1338, 1G2_2886, 1G2_2889, and 1G2_2902) exhibit potencies comparable to or greater than those of the small molecule inhibitor 1G2 [18]. In GC, IL-6 is significantly expressed and affects survival. In GC, M2 macrophage infiltration is prevalent and promotes communication between GES and IL-6. In vitro, GES-1 cells and THP-1-derived macrophages (a type of human macrophage generated from the THP-1 monocytic leukemia cell line) express IL-6 in response to stimulation. Furthermore, these cell lines can increase each other’s IL-6 levels in an autocrine manner, a process that is accelerated by *H. pylori* stimulation. Therefore, intercellular communication in the TME is dependent on IL-6. [19]. The expression of the natural killer group 2, member D (NKG2D) receptor was downregulated in biopsies of patients with *H. pylori* gastritis, although NKG2D ligands accumulated in the lamina propria, indicating NKG2D evasion. Specific *H. pylori* virulence factors are linked to the transcription and proteolytic shedding of NKG2D ligands in vitro by *H. pylori* in stomach epithelial cells. *H. pylori*-induced release of soluble NKG2D ligands diminished the cytotoxic activity of effector immune cells, particularly the antitumor activity of natural killer (NK) cells, and decreased the immunogenic visibility of infected cells. Therefore, a key mechanism for the activation of cytotoxic and NK T cells, which are crucial for tumor immunosurveillance and mucosal immunity, is the NKG2D receptor and its ligands. In addition to promoting GC growth, *H. pylori* may affect the NKG2D system, enabling altered cells to elude immune identification and progress uncontrolled to overt metastasis [20].

*H. pylori*-positive patients had significantly longer immune-related progression-free survival (irPFS) than *H. pylori*-negative patients (6.97 months versus 5.03 months, *p* < 0.001, hazard ratio [HR] 0.76, 95% confidence interval [CI] 0.62–0.95, *p* = 0.015) among 636 patients with Epstein–Barr virus-negative microsatellite-stable GC who were treated with anti-programmed cell death protein 1 (PD-1)/programmed death-ligand 1 (PD-L1) therapy. Furthermore, compared to the *H. pylori*-negative group, the *H. pylori*-positive group showed a trend toward a 4-month longer median immune-related overall survival (irOS). A “hot” TME was indicated by greater densities of PD-L1+ cells and nonexhausted CD8+ T cells in *H. pylori*-positive GC [21].

## 3. Variable Genes and Hosts

Genetics of host protection against *H. pylori* infections has been intensively studied in the medical literature (Table 2). By phosphorylating and nonphosphorylating various cellular pathways, including the mitogen-activated protein kinase (MAPK), phosphatidylinositol-3-kinase (PI3K)/protein kinase B (Akt), NF-κB, canonical Wnt/β-catenin (Wnt/β-catenin), Janus kinase (JAK)-signal transducer and activator of transcription proteins (STATs), and Salvador–Warts–Hippo (Hippo) signaling pathways, CagA can cause dysregulation when it enters the GEC. The shape, adhesion, polarity, proliferation, mobility, and other pathogenic alterations in GECs caused by these disrupted pathways ultimately contribute to the development of GC [22]. CagA is the main method by which *H. pylori* exhibits bacterial pathogenicity. Through phosphatidylinositol-3-kinase (PI3K)/protein kinase B (Akt) pathway activation, CagA suppresses forkhead box O3 (FOXO3a) transcription and translocates it from the nucleus to the cytoplasm [23].

*H. pylori* promoted cell migration and proliferation, boosted the inflammatory response, and inhibited apoptosis, which led to enhanced circPGD expression in infected human cells and three mechanisms that promoted GC spread. This finding suggested that *H. pylori*-related GC is influenced by CircPGD [24]. The formation of GCs is influenced by N6-methyladenosine (m6A) alterations and long noncoding RNAs (lncRNAs). Through interferon regulatory factor 2 (IRF2)-mediated transcriptional regulation and methyltransferase 14, N6-adenosine-methyltransferase subunit (METTL14)-mediated glycoprotein m6A alteration, *H. pylori* infection reduced the expression of lnc-(phospholipase C beta 1) PLCB1 [25]. The relationship between lncRNAs and *H. pylori* pathogenicity in GC has been the subject of an increasing amount of research. By decreasing the phosphorylation level of NF-Κb p65, long intergenic nonprotein-coding RNA 659 (LINC00659) knockdown dramatically reduced *H. pylori*-induced GES cell senescence and repressed IL-6 and IL-8 secretion [26]. m6A is the most frequent modification of mRNA that is regulated during *H. pylori* infection and is regulated by the lectin-like oxidized low-density lipoprotein (LDL) receptor-1 (LOX-1) receptor, a crucial factor for the adherence of *H. pylori* to the gastric epithelium. By downregulating LOX-1, which ultimately leads to decreased *H. pylori* adherence, the m6A alteration of host LOX-1 helps protect against *H. pylori* infection [27]. In GC clinical tissues, the expression of the transcription factor Snail1, which mediates the epithelial–mesenchymal transition, was positively linked with ubiquitin-specific protease 35 (USP35). Infection with *H. pylori* elevated USP35 and Snail Family Transcriptional Repressor 1 (Snail1) expression levels [28]. In terms of treatment, in GC linked or not to *H. pylori*, the proteins nuclear receptor corepressor 1 (NCOR1), KIT proto-oncogene, receptor tyrosine kinase (KIT), KRR1 small subunit processome component homolog (KRR1), and ESF1 nucleolar pre-RRNA processing protein homolog (ESF1) play significant roles in the molecular processes of S-1, docetaxel, and cisplatin [29]. An outer membrane adhesin is encoded by the outer inflammatory protein A (oipA) gene, which is linked to severe gastropathies, including GC. The severity of gastroduodenal illnesses was inversely correlated with the *H. pylori* oipA genotype (OR = 0.247, 95% CI: 0.0804–0.7149, and *p* = 0.007) [30].

From the point of view of genetics, one important risk factor is heredity. Lynch syndrome (LS) increases the risk of GC, along with other cancers. In terms of infection with *H. pylori*, surveillance with upper endoscopy has been conducted on genetically affected patients with this syndrome. Among those, only 6.7% of the cohort’s universally biopsied LS patients had either *H. pylori* or gastric intestinal metaplasia (5.5% with gastric intestinal metaplasia, 3.6% with *H. pylori*) [31].

One protective factor is natural flavonoids, such as apigenin, or Api, which are present in a variety of fruits and vegetables, including bell pepper, celery, parsley, garlic, and chamomile tea. By decreasing *H. pylori* colonization levels, inhibiting NF-κB activation, and reducing the generation of reactive oxygen species (ROS), Api decreases inflammation in *H. pylori*-infected cells [32].

Using biolayer interferometry, the link between *H. pylori*-glyceraldehyde 3-phosphate dehydrogenase (GAPDH) and several host compounds was examined. As GAPDH is known as a heme chaperone that participates in intracellular heme transport and engages molecular pathways, the relationship between *H. pylori* GAPDH and hemin was analyzed. Hemin has an affinity of 2.10 μM for *H. pylori* GAPDH, while *H. pylori* GAPDH bound to hemin is inactive. This indicates that hemin likely binds to the active site of *H. pylori* GAPDH to prevent substrate binding. Metal ions inhibit *H. pylori* GAPDH activity, indicating that they may also occupy substrate-binding sites [33].

Moreover, small ubiquitin-like modifier (SUMO) ylation is involved in the oncogenesis of GC through the activity of SUMO-activating enzymes. After evaluating the protein levels in GC patients, it was revealed that the overexpression of SUMO-activating enzyme subunit 1 (SAE1) is correlated with a worse outcome. In addition, the presence of *H. pylori* affects the expression of proteins by upregulating their levels [34].

To comprehend the genetic variety and pathogenicity of *H. pylori* in various populations and geographic locations, the predominant genotypes globally, as well as those found in Latin America and Chile, are discussed. The significant degree of sequence similarity observed in the virulence genes of *H. pylori* strains found in Spanish and Peruvian locals in Latin America points to a European impact. Geographically distributed strains that are more common in patients with more severe gastrointestinal disorders in Chile are strains that are positive for cagA and allelic variations of vacA s1 m1 [35]. HLA polymorphisms have a major impact on proper host cell immunological responses for the development of several diseases, including GC. In certain groups at risk, the analysis revealed that some of the polymorphisms are linked to *H. pylori*-related GC development and cancer-related death, but no correlation was found between the variability of the study population and the types of HLA polymorphisms studied [36]. Single nucleotide polymorphisms (SNPs) in *H. pylori* infection and Mammalian sterile 20-like kinase 1 (MST1) along with MST2 were examined for their potential correlation with the incidence of noncardia GCs. Haplotypes did not correlate with the incidence of noncardia GCs or *H. pylori* infection. The TT genotype (a homozygous dominant genotype) of rs7827435 and the CC genotype (a specific combination of alleles at a genetic locus) of rs10955176 may be risk factors for noncardia GC and *H. pylori* infection, respectively [37].

The stomachs of more than half of the world’s population are colonized by the spiral gram-negative bacterium *H. pylori*, which can lead to peptic ulcers, GC, MALT lymphoma, and other gastroduodenal illnesses. The complicated interplay between virulence factors and phase-variable genes produced by the infecting strain and the host genetic background affects the clinical results of *H. pylori* infection [38].

## 4. CG Clinical Trajectory in *H. pylori*-Eradicated Patients

Eliminating *H. pylori* has no negative impact on GCs’ clinical trajectory [39]. When invasive tumors are discovered more than ten years after *H. pylori* eradication, their histological type and pathological stage are likely more aggressive [40]. The eradication of *H. pylori* only decreases the incidence of GC by 46% because of preexisting precancerous lesions [41]. For GC, there were decreasing trends in the population attributable to fractions of smoking, pickled vegetables, alcohol intake, and *H. pylori* infection but increasing trends in the population attributable to fractions of diabetes and unhealthy body mass index. However, to lessen the burden of this extremely dangerous cancer in the future, measures for controlling risk factors must continue to be effective [42]. In certain groups, people with upper gastrointestinal symptoms frequently have *H. pylori* infection, which is linked to intestinal metaplasia. These findings indicate that intestinal metaplasia can be positively predicted by age and *H. pylori* infection status, underscoring the significance of promptly detecting and treating *H. pylori* infection [43]. Women frequently develop GC, often at an advanced stage, with a poor prognosis. Female GC patients have comparatively low survival rates, which emphasizes the importance of preventative measures such as thorough *H. pylori* eradication [44].

## 5. Treatment

The medical literature examines the current options for treating *H. pylori* infections in clinical practice (Table 3). A total of 53 of the 82 individuals with *H. pylori* infection underwent eradication therapy. Of these, 26 (49%) had a success rate of only 38% (10/26) according to eradication tests after treatment. Low patient adherence to the follow-up plan necessitates future reinforcement to guarantee better clinical care. [45]. Using the qPCR method, 132 people tested positive for the *H. pylori* ureA gene. The highest recorded rates of resistance were 82.6% (n = 109) for clarithromycin (CLA), 69.7% (n = 92) for levofloxacin (LFX), and 62.1% (n = 82) for dual resistance. Antibiotic susceptibility testing or a regional profile of antibiotic resistance should be used to guide treatment decisions for individuals who have previously failed therapies [46]. Eradication therapy for *H. pylori* has substantial hurdles. The effective treatment of this difficult infection depends on continued study in this area and the ongoing improvement in therapeutic options [47].

The PubMed database currently has almost 50,000 publications from the 40 years that have passed since H. pylori was discovered, demonstrating the volume and quality of the studies conducted on the significance of this bacterium. Current clinical research focuses mostly on developing strategies to prevent GC and developing treatments to combat the increase in antibiotic resistance [48]. The outcome of treatment for GC has not changed appreciably for more than ten years. Methionine-restricted therapies have been shown to inhibit GC through metabolic reprogramming, indicating their potential as adjuvant treatments for this disease [49]. A greater incidence of GC was strongly correlated with longer potassium-competitive acid blocker (PCAB) use and high-dose PCAB use. According to sensitivity studies, compared to proton pump inhibitor users (PPIs), PCAB users had a similar risk of GC incidence [50]. More than 90% eradication effectiveness is typically attained with triple therapy based on bismuth and non-bismuth. The same recommendations validated for the adult population also apply to older patients to achieve optimal *H. pylori* eradication effectiveness: avoiding prescriptions for repeated treatments, selecting quadruple therapies, prescribing longer treatment durations, and prescribing high-dose proton pump inhibitors twice daily [51]. An antimicrobial peptide (AMP) hydrogel offered two advantages over traditional antibiotic treatments: it eliminated the need for PPIs throughout the treatment and quickly killed germs in the gastric juice. Antibiotics and AMP hydrogels inhibited the production of proinflammatory cytokines, but only the former enhanced the resolution of inflammation [52]. Vonoprazan, also known as Voquezna^®^, is a powerful and long-lasting acid blocker that is a first-class potassium-competitive drug. Adults with *H. pylori* infection can now receive approval in the USA to use amoxicillin-vonoprazan dual therapy [53]. Novel approaches were devised to efficiently resolve and precisely quantify the overlapping spectra of amoxicillin and vonoprazan. The concentrations of vonoprazan were 3–60 μg mL^−1^, and the concentrations of amoxicillin were 5–140 μg mL^−1^ [54].

Interventions such as the use of probiotics, prebiotics, or antibiotics can alter the gut microbiota. GC is associated with dysbiosis and immunological dysregulation caused by alterations in the normal gut microbiota [55]. Traditional antibiotic therapy in conjunction with probiotics and postbiotics is currently receiving much attention for the treatment of *H. pylori* infection. Treatment with *Lactobacillus crispatus*-derived extracellular vesicles (EVs) and cell-free supernatant (CFS) significantly reduced *H. pylori*-induced IL-8 production [56]. The degree of immune protection attained by a possible nanovaccine prepared for oral vaccination against *H. pylori* infection, alginate/pCI-neo-UreH, was comparable to what was observed when formalin-killed *H. pylori* Hel 305, a commonly used positive control, was used. Strong intestinal and systemic Th1 responses, as well as notable mucosal and systemic antigen-specific antibody responses, were induced by the alginate/pCI-neo-UreH nanovaccine composition [57].

Although the difference was not statistically significant, there was a trend toward greater preventive effectiveness with high-dose Moluodan. Compared with the folic acid group (53.9%; *p* = 0.006), the Moluodan group had a greater rate of dysplasia elimination (82.8%). There were no significant side effects linked to the medication [58]. Ferroptosis, which is linked to fatal lipid peroxidation, is crucial for both viral and malignant illnesses. In vitro and in vivo, ferroptosis sensitivity is conferred on GC cells by CagA. CagA increases the expression of alkylglycerone phosphate synthase (AGPS) and 1-acylglycerol-3-phosphate O-acyltransferase 3 (AGPAT3), which stimulates the synthesis of polyunsaturated ether phospholipids (PUFA-ePLs) and increases the risk of ferroptosis. CagA-positive cells are more responsive to apatinib than CagA-negative cells [59].

It was proposed that the Hezi Qingyou Formula (HZQYF), which contains cloves, Ficus hirta Vahl, and Chebulae fructus, could have anti-*H. pylori* activity. The composition of the formula may alter the morphology of *H. pylori*, increase the permeability of its extracellular membrane, decrease the activity of urease, and downregulate the expression of adhesion genes (alpA, alpB, and babA), urease genes (ureE and ureF), and flagellar genes (flaA and flaB) [60].

## 6. Future Directions

By using nanoparticles for targeted drug administration, nanomedicine provides a viable way to combat delivery problems and antibiotic resistance. Because of the properties of the stomach mucosa, nanocarriers, including liposomes and nanoparticles, improve medical stability and circulation while focusing on infection sites [61]. The main determinant of gastric oncogenetics and the main cause of environmental gastritis is *H. pylori*. It is necessary to encourage additional progress in the clinical management of gastritis patients. The following crucial RE.GA.IN consensus statements should be noted: 1. Depending on variables such as etiology and stage, there are varying degrees of risk associated with the preneoplastic condition known as gastritis. 2. *H. pylori* is still the most common cause of chronic gastritis worldwide, but it is becoming less common worldwide, especially in younger people. 3. It is unknown how additional factors, in the absence of *H. pylori* or after its eradication, contribute to the epidemiology of chronic gastritis [62]. It has also been demonstrated that a fraction of *H. pylori*-infected children, especially those harboring virulent strains, present premalignant lesions. Consequently, the earliest premalignant gastrointestinal alterations may occur in childhood, providing a promising avenue for future research to determine the exact border between infection and cancer [63].

## 7. Conclusions

Numerous studies have been conducted on *H. pylori*/GC due to the significant fatality rate associated with the illness. Numerous fields, such as immunopathology, genetics, diagnostics, and treatment, are the focus of research. Research of this kind needs to be prioritized.

## Figures and Tables

**Table 1 cancers-16-01958-t001:** Major developments in immunopathology studies of *H. pylori* infection.

Immunological Studies	References
The development of novel GC treatment targets may be influenced by the tumor immune microenvironment (TIME) of GC.	[14]
TLR6 might be a good therapeutic candidate.	[15]
FOXP3+ T lymphocytes pick up on triggers related to inflammation.	[16]
DNA vaccines against *H. pylori* pIRES2-DsRed-Express-ureF may be useful as immunotherapeutics.	[17]
The inhibition of T4SS-determined extracellular pili assembly by 1G2 and its derivatives provides a mechanism for their anti-virulence impact.	[18]
IL-6-driven autocrine and paracrine positive feedback loops between macrophages and stomach epithelial cells.	[19]
*H. pylori* changes the NKG2D pathway.	[20]
Tumor microenvironments are heated by *H. pylori* infection, and this outcome is beneficial for immunotherapy in the treatment of GC.	[21]

**Table 2 cancers-16-01958-t002:** Major developments in variable genes and host studies of *H. pylori*.

Genetics Studies	References
GC cells’ transition into a stemness stage and the regulating role of the Akt/FOXO3a axis.	[23]
For therapeutic intervention, CircPGD might be a promising new target.	[24]
One potential new target for GC therapy is lnc-PLCB1.	[25]
For stomach disorders linked to *H. pylori*, LINC00659 may prove to be a unique, promising prognostic and therapeutic marker.	[26]
*H. pylori* adhesion is decreased by the m6A alteration of the host LOX-1 mRNA.	[27]
One possible target for GC treatment could be USP35.	[28]
The proteins NCOR1, KIT, MITF, ESF1, ARNT2, TCF7L2, and KRR1 have been linked to GC, whether or not *H. pylori* is present.	[29]
An essential pathogenicity marker, the *H. pylori* oipA genotype, has been linked to the severity of gastric disorders.	[30]
Significance of baseline biopsies of the stomach antrum.	[31]
Api controls several GC hallmarks.	[32]
GAPDH has been identified as a heme chaperone.	[33]
SEA1 may be a potential treatment option.	[34]
Genetic variety of *H. pylori* in various parts of the world.	[35]
The development of GC and mortality related to *H. pylori* are linked to particular polymorphisms.	[36]
The TT genotype of rs7827435 and the CC genotype of rs10955176 may operate as protective factors against the risk of noncardia GC and *H. pylori* infection, respectively.	[37]
The course of an infection is determined by the interplay among virulence factors, phase-variable genes, and host genetics.	[38]

**Table 3 cancers-16-01958-t003:** Major developments in the treatment of *H. pylori*.

Current Treatment Options Studies	References
Lower eradication rate with the empirical regimens employed.	[45]
High secondary resistance rates to LFX and CLA in treatment.	[46]
Antibiotic- and multidrug-resistant forms are becoming more common.	[47]
Gastritis caused by *H. pylori* is classified as an infectious condition, and all symptomatic patients need to receive treatment.	[48]
Methionine-restricted therapies have been shown to inhibit GC.	[49]
In *H. pylori*-eradicated individuals, the use of PCABs was linked to a greater risk of GC, with effects according to dosage and duration.	[50]
Optimal eradication effectiveness (>90%) is mostly achieved with bismuth- and non-bismuth-based quadruple therapies.	[51]
Effective and biosafe AMP hydrogels could be good options for the clinical management of *H. pylori* infections.	[52]
In PHALCON-HP, the dual- and triple-therapy regimens based on vonoprazan were typically well tolerated.	[53]
Pharmacological combinations such as vonoprazan and amoxicillin have synergistic results.	[54]
Intake of probiotics and prebiotics is being researched to enhance therapeutic results and lessen side effects related to anticancer medication.	[55]
The EVs and CFS of the *L. crispatus* strain RIGLD-1 are possible therapeutic agents to combat inflammation caused by *H. pylori.*	[56]
One possible nanovaccine to be used in an oral vaccination against *H. pylori* infection is alginate/pCI-neo-UreH.	[57]
One pack of Moluodan, taken three times a day for a year, was a safe and effective way to reverse stomach precancerous lesions, particularly dysplasia, in individuals who tested negative for *H. pylori*. An improvement in efficacy was observed when the dosage was doubled.	[58]
A patient’s CagA status for *H. pylori* may help stratify patients susceptible to apatinib treatment.	[59]
HZQYF showed potential anti-*H. pylori* activity in vitro by downregulating the expression of genes related to flagellar adhesion, urease, and bacterial adhesion.	[60]

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
