# Peer review of "Helicobacter pylori Infection in Patients with Gastric Cancer: A 2024 Update"

_cancers, 2024, doi:10.3390/cancers16111958_

Round 1
Reviewer 1 Report
Comments and Suggestions for Authors
Manuscript deal with important problem Helicobacter pylori infection as etiological factor of gastric cancer and review paper published in 2024. It is especially important to destribute this data among oncologists.
I would recommend include in to paper wery important statements from REGAIN consensus, which alredy cited in the manuscript. Gastritis is a preneoplastic condition with different levels of risk, depending on factors such as aetiology and stage. Worldwide, H. pylori is still the leading known cause of chronic gastritis; its prevalence is declining in many parts of the world, particularly in young populations. The contribution of other factors to the epidemiology of chronic gastritis in the absence of H. pylori or after its eradication remains to be determined.
Author Response
We thank you for your appreciation and valuable comments. We introduced statements from the RE.GA.IN consensus in our article.
Reviewer 2 Report
Comments and Suggestions for Authors
1. The Abstract section is too simple, making it difficult for readers to have an overview of the entire text. The author should make revisions. 。
2. The "Models for Studying H. pylori Infection" section does not introduce models related to gastric cancer research. The mouse infection model does not reflect the protective efficacy of vaccines in the population. Suggest making modifications based on the content of the topic.
3. In line134-135:“Cancer-causing genes are present in 50% of native H. pylori strains, and they also 134 exhibit hypervirulence.” is not applicable to multiple Asian countries. It is recommended to add a differential analysis of the structure of CagA, especially the phosphorylation sites.
4. Part of Line 217-223 and all of contents of "5. Diagnosis" (lines 124-158) deviates from the scope of this review and is recommended to be deleted.
5. Suggest moving the "line 276-286" section to the "Treatment" section.
Author Response
1. We have redone the abstract;
2. The "Models for Studying H. pylori Infection" section was deleted from our article. We focused on the purpose of the study, which was to present the major discoveries of the year 2024.
3. We deleted this sentence. "Cancer-causing genes are present in 50% of native H. pylori strains, and they also exhibit hypervirulence." And we explained the analysis of the structure of CagA, especially the phosphorylation sites, from another bibliographic reference.
4. Part of Line 217-223 and all of the contents of "5. Diagnosis" (lines 124–158) have been deleted from our article. We focused on the purpose of the study, which was to present the major discoveries of the year 2024.
5. We moved the "line 276-286" section to the "Treatment" section.
Round 2
Reviewer 2 Report
Comments and Suggestions for Authors
The author has made detailed revisions to the review comments, resulting in a significant improvement in overall quality and readability of the article. The article basically meets the publishing requirements.